# Genomic Analysis of *Clostridioides difficile* Recovered from Horses in Western Australia

**DOI:** 10.3390/microorganisms11071743

**Published:** 2023-07-03

**Authors:** Natasza M. R. Hain-Saunders, Daniel R. Knight, Mieghan Bruce, David Byrne, Thomas V. Riley

**Affiliations:** 1Centre for Biosecurity, and One Health, Harry Butler Institute, Murdoch University, Murdoch, WA 6150, Australia; 30089918@student.murdoch.edu.au (N.M.R.H.-S.); mieghan.bruce@murdoch.edu.au (M.B.); 2School of Biomedical Sciences, The University of Western Australia, Queen Elizabeth II Medical Centre, Nedlands, WA 6009, Australia; daniel.knight@health.wa.gov.au; 3PathWest Laboratory Medicine, Department of Microbiology, Queen Elizabeth II Medical Centre, Nedlands, WA 6009, Australia; 4School of Veterinary Medicine, Murdoch University, Murdoch, WA 6150, Australia; d.byrne@murdoch.edu.au; 5School of Medical and Health Sciences, Edith Cowan University, Joondalup, WA 6027, Australia

**Keywords:** *Clostridioides difficile*, equine, one health, genomics

## Abstract

*Clostridioides difficile* poses an ongoing threat as a cause of gastrointestinal disease in humans and animals. Traditionally considered a human healthcare-related disease, increases in community-associated *C. difficile* infection (CDI) and growing evidence of inter-species transmission suggest a wider perspective is required for CDI control. In horses, *C. difficile* is a major cause of diarrhoea and life-threatening colitis. This study aimed to better understand the epidemiology of CDI in Australian horses and provide insights into the relationships between horse, human and environmental strains. A total of 752 faecal samples from 387 Western Australian horses were collected. *C. difficile* was isolated from 104 (30.9%) horses without gastrointestinal signs and 19 (37.8%) with gastrointestinal signs. Of these, 68 (55.3%) harboured one or more toxigenic strains, including *C. difficile* PCR ribotypes (RTs) 012 (*n* = 14), 014/020 (*n* = 10) and 087 (*n* = 7), all prominent in human infection. Whole-genome analysis of 45 strains identified a phylogenetic cluster of 10 closely related *C. difficile* RT 012 strains of equine, human and environmental origin (0–62 SNP differences; average 23), indicating recent shared ancestry. Evidence of possible clonal inter-species transmission or common-source exposure was identified for a subgroup of three horse and one human isolates, highlighting the need for a One Health approach to *C. difficile* surveillance.

## 1. Introduction

*Clostridioides difficile* poses an ongoing threat as a cause of diarrhoea and gastrointestinal disease in animal and human populations. Although historically considered a healthcare-related disease, a distinct increase in community-associated human *C. difficile* infection (CDI) and the emergence of human disease caused by apparently animal strains has sparked a growing interest in CDI being a zoonotic infection and the pathways of transmission of *C. difficile* in the community [1,2]. To compound this issue, in 2019, escalating antimicrobial resistance (AMR) in *C. difficile* resulted in the USA Centers for Disease Control and Prevention (CDC) listing it as one of the top five microorganisms posing an urgent AMR threat to public health in the USA [3]. This suggests a need for both antimicrobial stewardship and a “One Health” approach to disease investigation, prevention and control.

*C. difficile* is an anaerobic Gram-positive bacillus, transmitted through the faecal–oral route and resident in the intestinal tracts of both human and non-human animals [4]. The ability to form hardy spores allows persistence in a diverse range of environments. Disease is toxin-mediated, through the disruption of cell integrity, cell signal interruption and the subsequent induction of inflammatory mediators by toxins A (TcdA) and B (TcdB) [5,6,7]. The extent to which a third binary toxin (CDT—an actin-ADP-ribosylating protein) impacts disease remains controversial, although reports of increased pathogenicity have gained momentum [8].

Carriage of *C. difficile* has been reported widely in mammals, birds, reptiles and insect species, within both wild and domestic populations [9]. Its high prevalence in production and companion animals has raised concerns about reservoirs, with paired pig/farmer and pet/owner studies with molecular comparative analysis confirming the cross-species relatedness of *C. difficile* strains [10,11,12,13].

In horses, *C. difficile* is recognised as a major cause of enteric disease [14]. Disease manifestation can span from simple gastrointestinal (GI) discomfort or diarrhoea to pseudomembranous colitis with fatal complications. Asymptomatic carriage resulting in the subclinical shedding of spores is also well documented and spores survive in inoculated horse faeces for up to 4 years [15]. Globally, the isolation rates for *C. difficile* in horses remain highly variable, with differences between diarrhetic animals (adults 0–90%; foals 0–33%) and those without GI signs (adults 0–25%; foals 0–44%). However, transient colonisation and distinct regional differences appear to be common [4,15,16,17,18,19]. 

In Australia, the potential impact of equine CDI is three-fold, encompassing animal welfare, socioeconomics and One Health concerns. Horses are well integrated into the Australian lifestyle, utilised as companion, performance and production animals. Horse racing and breeding within the country represent a multi-billion dollar industry and, although dwindling, Australia remains one of the top horse meat producers in the world [20,21]. The implications of reduced productivity due to *C. difficile*-associated disease, as well as health concerns for the animal, are therefore considerable. 

*C. difficile* disease prevention and control in general, and specifically for horses, has been plagued by difficulties. A combination of a complicated bacterium–host interaction, poor predictability of host outcome, an absence of standardised testing in horses and an inherent lack of surveillance creates multiple problems. Many questions remain regarding the extent of this issue in Australian equine populations, the impact of asymptomatic carriage and the relationship of these two factors with the environment and human disease, implying that a One Health approach is required. This study aimed to bridge these knowledge gaps by using a multifaceted approach to investigate the epidemiology of CDI in domestic and hospitalised horse populations and assess genetic comparability with existing human and environmental databases.

## 2. Materials and Methods

### 2.1. Study Population and Sample Collection

A total of 752 faecal samples from 387 Western Australian (WA) domestic horses were collected from privately owned companion animals, stud farms, racing training centres and horses admitted to The Animal Hospital at Murdoch University (TAHMU), Equine Services. These convenience samples were collected between June 2020 and June 2022.

For the stud farm, racing and companion animals, a single faecal sample was collected. Environmental samples were also taken close to where the faecal samples were collected to assess for potential contamination. Equine hospital samples were obtained periodically from all horses present at the clinic at the time of sampling over the 2-year period. For all subjects, samples were collected fresh and kept refrigerated at 4 °C until processing.

Horses were categorised based on GI status. Those with diarrhoea, colitis or colic (signs of abdominal pain) were considered “GI symptomatic”.

### 2.2. Case Study Samples

MU4: A 5-year-old thoroughbred stallion was admitted to hospital, presenting with acute onset diarrhoea and colic, consistent with colitis. Ultrasound showed fluid in the colon and distension in the small intestine. Pathology identified leukocytosis, hypoproteinaemia, hypovolemic shock and a marked electrolyte loss. The patient was placed in isolation and treated for suspected infectious bacterial colitis with gentamicin, penicillin, metronidazole, enteral di-tri-octahedral smectite powder and sodium bicarbonate. Additional medication for pain (flunixin) and anticoagulation (dalteparin), as well as plasma and electrolyte replacement, was also administered. Ice boots for the prevention of laminitis were fitted and replaced at 2 h intervals. The horse showed nominal improvement in response to treatment.

PCR testing on a faecal sample later confirmed the presence of *C. difficile* toxin A and toxin B genes with faecal transfaunation being carried out on day 5 of hospitalisation for additional treatment. On day 8 of hospitalisation, complete ischaemia of the distal limb had developed and euthanasia on welfare grounds was conducted. Postmortem examination revealed diffusely thickened and necrotic mucosa of the dorsal colon, as well as haemorrhage within the submucosa and tunica muscularis layers, consistent with *C. difficile* colitis.

MU5: A second associated 3-year-old gelding was admitted to hospital with diarrhoea, marked neutrophilia and pyrexia. Ultrasound showed the stomach to be gas- and fluid-filled. The patient was placed in isolation and treated for suspected infectious bacterial colitis with metronidazole, enteral di-tri-octahedral smectite powder and sodium bicarbonate. Flunixin was administered for pain, along with dalteparin and ice boots as preventative measures. Faecal samples tested positive for *C. difficile* glutamate dehydrogenase antigen and toxins. The patient responded well to treatment and was released after 4 days of hospitalisation.

Both horses had travelled long distances in the week prior to manifestation of clinical signs and had been stabled adjacent to a third horse (which was the first to show signs of diarrhoea, chronologically). This additional horse was not seen at the treating hospital. 

### 2.3. Culture

Faecal samples were cultured for *C. difficile* using previously described methods [22] with some changes. For solid faeces, individual pellets were carefully cut on the transverse plane and a portion extracted from the core to avoid surface environmental contamination. For liquid faeces, samples were mixed and an aliquot was taken using a sterile transfer pipette. A 10 mL sample was added to a 90 mL brain heart infusion selective enrichment broth supplemented with 5 g/L yeast extract, 1 g/L l-cysteine, 1 g/L taurocholic acid, 250 mg/L cycloserine and 8 mg/L cefoxitin (BHIB-S) (PathWest Media, Mt Claremont, WA) and incubated with a loose lid for 7 days in an A35 anaerobic chamber (Don Whitley Scientific Ltd., Yorkshire, UK) at 35 °C with an atmosphere of 80% nitrogen, 10% hydrogen and 10% carbon dioxide, and a relative humidity of 75%. Following incubation, the BHIB-S cultures were alcohol-shocked to select for spores and a 10 µL aliquot was plated onto ChromID^®^ *C. difficile* agar (bioMérieux, Marcy-l’Étoile, France), which was incubated anaerobically for 48 h. Putative *C. difficile* colonies were sub-cultured onto blood agar and incubated for up to 48 h. *C. difficile* was confirmed by characteristic ground-glass colony appearance, distinct barnyard odour and chartreuse fluorescence under UV light (~360 nm) [23].

### 2.4. C. difficile Characterisation

Isolates were characterised by ribotyping and toxin gene profiling using multiplex PCR and capillary electrophoresis as previously described [24]. Toxin gene profiles were determined by the presence of *tcdA* (toxin A), *tcdB* (toxin B) and *cdtA* and *cdtB* (binary toxin). For ribotype (RT), banding patterns were visualised using QIAxcel ScreenGel software (QIAgen, Venlo, The Netherlands). Densitometric curve and band pattern analysis was performed using the Bionumerics v7.1 software package (Applied Maths, Sint-Martens-Latem, Belgium) based on Dice coefficient and Pearson correlation and comparison with a database of over 1000 strains of both internationally recognised and local strains.

### 2.5. Antimicrobial Susceptibility Testing (AST)

Minimum inhibitory concentrations (MICs) of 10 antimicrobial agents were determined for all *C. difficile* isolated, using the agar incorporation method described by the Clinical and Laboratory Standards Institute (CLSI) [25]. The antimicrobials fidaxomicin, vancomycin, metronidazole, rifaximin, clindamycin, erythromycin, amoxicillin-clavulanate, moxifloxacin, meropenem and tetracycline were chosen for testing based on their usage in veterinary practice, human healthcare and CDI treatment and prevention. Clinical breakpoints used for vancomycin and metronidazole were recommended by the European Committee on Antimicrobial Susceptibility Testing (EUCAST) [26], while those for fidaxomicin and rifaximin were proposed by the European Medicines Agency (EMA) [27] and O’Connor et al. [28]. All other breakpoints were based on CLSI recommendations [29]. MIC_20_, MIC_50_, MIC_90_ and geometric mean (GM) MIC values were calculated for all antimicrobials.

### 2.6. Whole-Genome Sequencing (WGS) and Analysis

DNA extraction was performed on a subset of 28 equine *C. difficile* strains of interest using the QuickGene DNA DT-S tissue kit as per the manufacturer’s instructions (Kurabo Industries, Osaka, Japan). Short-read genome sequencing was performed on an illumina NovaSeq 6000 platform, with Nextera Flex 150 bp paired-end chemistry to a depth of 145X coverage (illumina, San Diego, CA, USA). Long-read sequencing was performed on the Nanopore MinION Mk1C platform, utilising the sequencing libraries prepared using the Ligation Sequencing Kit (SQK-LSK109) and the same gDNA extract as above, and run on a FLO-MIN106 flow cell for 12 h (Oxford Nanopore Technologies, Oxford, UK). The multi-locus sequence type (ST) of strains was determined using PubMLST, based on the comparison of seven known housekeeping genes (BIGSdb v1.32.0) [30]. Abricate v1.0 (github.com/tseemann/abricate, accessed on 14 June 2022) was used to screen genomes against the NCBI database for horizontally acquired AMR genes and against a custom tcdA/B/CDT gene list based on information from the NCBI database for *C. difficile* toxins. Additional genomic features were identified by interrogation against the NCBI database through BLAST [31].

### 2.7. ST54 Whole-Genome SNP Phylogeny

A subset of 6 equine *C. difficile* RT 012 (ST54) strains along with 2 environmental and 37 Australian human RT 012 strains from our laboratory collection (unpublished) were selected for further interrogation. Split k-mer analysis was performed using SKA v1.0 [32]. Pairwise SNP differences between strains were calculated using snp-dists v0.8.2 (https://github.com/tseemann/snp-dists, accessed on 21 June 2022). Construction of a maximum-likelihood tree was based on alignment of polymorphic sites visualized using iTOL v6.0 [33].

### 2.8. Hybrid Assembly of Equine C. difficile RT 012 Closed Reference Genome

Equine *C. difficile* isolate, MU4, was selected for hybrid assembly, after being cultured from a confirmed case of symptomatic CDI as described above. Short and long sequence reads were trimmed with TrimGalore v0.6.7 (https://github.com/FelixKrueger/TrimGalore, Cambridge, UK, accessed on 10 June 2022). Long reads were additionally filtered using Filtlong v0.2.0 (https://github.com/rrwick/Filtlong, accessed on 10 June 2022) to remove low-quality reads of less than 1000 base pairs. Hybrid assembly of short- and long-read data was then performed utilising the Unicycler v0.5.0 pipeline in conservative assembly mode [34] with multiple rounds of polishing (pilon v1.2.4, racon v1.4.3) to improve contiguity. The final assembly graph was visualised with Bandage v0.9.0 [35]. Genome annotation was performed by submission to NCBI using the Genome Annotation Pipeline v5.2 [36] with the complete genome assembly data submitted to GenBank under BioProject PRJNA865075 (accessions CP102397, CP102398 and CP102399). Prophages were characterised using PHASTER [37]. 

### 2.9. Statistical Analysis

Comparisons of proportions were performed using Chi-squared test, with a two-tailed *p*-value < 0.05 considered statistically significant.

## 3. Results

### 3.1. Epidemiology

In total, 148 strains of *C. difficile* were isolated from 123 horses across all study locations. A summary of the overall proportions positive for *C. difficile* in various groups of horses is outlined in Table 1. *C. difficile* was identified in 19 (37.8%) horses with GI signs and 104 (30.9%) without GI signs. There was a higher recovery from *C. difficile* in foals in both the GI-signs (60%) and non-GI-signs (46.7%) groups compared to adult horses (34.8% and 30.2%, respectively), although this difference was not statistically significant. There was a significant difference between *C. difficile* isolation rates from non-GI-signs adult horses from the veterinary hospital population and those from the private domestic population (*p* = 0.026); however, this was not seen in the adult cohort with GI signs.

Of the 123 horses that *C. difficile* was isolated from, 68 (55.3%) harboured one or more toxigenic strains. In total, 95 of the 148 strains isolated (64.2%) contained *tcdA* and *tcdB* genes (A+B+), with 1 additional strain also possessing the binary toxin genes *cdtA* and *cdtB* (CDT+). 

A combination of novel (42.6%) and previously described (57.4%) *C. difficile* RTs were identified, including both internationally recognised strains (prefixed RT) and locally identified strains (prefixed QX). The most common strain was *C. difficile* RT 014/020 (*n* = 15), followed by RTs 012 (*n* = 9), 087 (*n* = 8), 010 (8) and 002 (*n* = 5), all of which have been previously isolated from humans. Figure 1 presents the toxin profiles and RTs of the strains isolated from all horses. 

For the private domestic horse sample group, *C. difficile* recovery varied considerably between individual sites, ranging from 0 to 78%. Figure 2 demonstrates the disparity between sites within this cohort. Differences were seen also between the horse-use classifications associated with these sites (companion animals 15.2%, stud farms 48.4% and racing 30.8%). Isolations from companion animal sites were significantly lower than those from racing training centres (*p* = 0.017) and stud farms (*p* = 0.0002). 

Within the equine hospital group, *C. difficile* recovery fluctuated throughout the duration of stay. In those horses that were sampled on more than one occasion, *C. difficile* was recovered consistently and samples of the same *C. difficile* RT were recovered in only 7 of 49 cases. Shedding of multiple strains over the sampling period was seen in 13 (6.9%) subjects, suggesting that both true colonisation and transient shedding existed simultaneously within this population. Figure 3 summarises the longitudinal data depicting the isolation of *C. difficile* from horses at the TAHMU Equine Services.

### 3.2. Antimicrobial Susceptibility Testing (AST)

Susceptibility to antimicrobials used to treat both animal and human CDI remained high. All strains were susceptible to the veterinary CDI antimicrobial of choice, metronidazole (MIC < 2 mg/L), while only one strain (non-toxigenic isolate MU99) was resistant to fidaxomicin (MIC > 1 mg/L) and two isolates (MU99 and MU207) were resistant to vancomycin (MIC > 2 mg/L). 

None of the strains showed resistance to meropenem or amoxicillin-clavulanate, while clindamycin had the largest number of resistant strains (*n* = 77), followed by erythromycin (*n* = 30), rifaximin (*n* = 18) and tetracycline (*n* = 15).

Multi-drug resistance (resistance to three or more antimicrobial classes) was seen in four strains, all of which were toxigenic *C. difficile* RT 012. This included isolates MU200, MU202, MU5 and MU4, the latter being implicated in the confirmed case of severe CDI mentioned earlier. All four strains showed resistance to rifaximin, clindamycin, erythromycin, moxifloxacin and tetracycline. 

Figure 4 presents the MIC distributions for the 10 antimicrobial agents tested against 160 equine strains isolated.

### 3.3. Molecular Characterisation

The 28 *C. difficile* strains of interest that were sequenced comprised eight STs, all of which belonged to evolutionary MLST clade 1. All RTs were concordant with the ST determined through WGS. The genomic toxin gene data identified for these samples were consistent with the toxin gene profiles determined by PCR. 

### 3.4. ST54 Whole-Genome SNP Phylogeny 

A maximum-likelihood tree of 45 *C. difficile* RT 012 (ST 54) strains isolated from equine (*n* = 6), environmental (*n* = 2) and human origins (*n* = 37) is shown in Figure 5. This SKA analysis identified a phylogenetic cluster of 10 closely related RT 012 (ST54) equine, human and environmental strains separated by 0 to 62 SNPs (average 23). A further two human isolates (N1QLD137 and STA0015) were, on average, 46 SNPs from the equine cluster, indicating recent shared ancestry. 

Of these, three horse isolates (MU200, MU201 and DH155) and one human isolate (STA1033) had a difference of two SNPs, indicating possible evidence of clonal inter-species transmission or common source exposure. The human strain was isolated from a WA patient in 2016, two of the equine isolates (MU200 and MU201) were recovered from horses hospitalised in WA in 2020 and 2021, while DH155 was from a diarrhetic foal residing at a stud farm. The direction of transmission could not be determined with the information collected.

### 3.5. Complete Genome for MU4

A complete and circularised genome for an RT 012 (ST 54) strain of *C. difficile* (MU4) was generated. The genome consisted of a single chromosome of 4,492,337 bp containing three prophages, as well as a novel extrachromosomal plasmid and phage. AMR genes for beta-lactams, vancomycin, macrolides, streptomycin, tetracycline, trimethoprim and amikacin/kanamycin were seen within the chromosome. No known AMR genes were identified within the extrachromosomal elements. Table 2 outlines these findings and the associated chromosomal assembly metrics. 

## 4. Discussion

Comprehensive epidemiological investigations are imperative before establishing effective CDI prevention and control policies in horses. The data presented here address a key knowledge gap about *C. difficile* in equine populations and provide a baseline for the understanding of *C. difficile* and CDI in horses in WA. The overall recovery of *C. difficile* from non-GI-sign adult horse faeces (30.9%) was higher than previously found in Australia and globally (0–14%) [4,15,16,17,18,19]. There was, however, considerable variation in *C. difficile* isolation between individual sites, which corroborates views that CDI is sporadic in nature with a tendency towards local outbreaks [14]. The lack of historic *C. difficile* data precludes any trend analysis and highlights the need for further studies.

A lack of standardisation in *C. difficile* culturing methods and ribotyping protocols and the absence of a comprehensive open access ribotype library have been cited previously as major limitations to carrying out effective surveillance studies to identify global patterns in emerging ribotypes [38,39]. In addition, many of the methods for detecting *C. difficile* have been developed for human samples, reflecting the historical perception of *C. difficile* causing just human healthcare-related disease with little concern for wider environmental sampling [40]. While attempts have been made in recent years to develop tests optimised for animal and environmental samples, reference methods are yet to be established [41,42].

In this study, increased volumes of BHIB (90 mL compared to 10 mL) and sample (10 g compared to 1–5 g) were used for culture compared to previous studies in horses to increase sensitivity and maximise the recovery of *C. difficile*. A non-homogenous distribution of spores within a sample, particularly in those with low spore numbers, can result in an underestimation of prevalence [43,44]. Given the theoretically low infectious dose required to initiate CDI, accuracy depends on sensitive culturing methods [45]. While *C. difficile* spore numbers and distribution in equine faeces are not known, evaluation of livestock and other ruminant faeces suggests uneven dispersal, the impact of which could be minimised with a larger sample volume/weight and multiple samples [43,44]. Although a direct comparison of various broth and sample volumes was not carried out in the current study, a pilot study did identify an increase in recovery with more sample. Regardless of external comparisons, such a high number of non-GI-sign individuals harbouring *C. difficile* is of significant concern for both infection control and the difficulty of diagnosing true disease.

To compound the issue, the inconsistent *C. difficile* shedding patterns seen within this study create further complexity for CDI laboratory diagnosis. In the hospital-based population, the detection of *C. difficile* occurred predominantly intermittently. Furthermore, longitudinal equine hospital samples showed shedding of multiple strains in individual horses in 13 (6.9%) subjects over the study period, suggesting both true colonisation and transient shedding exist simultaneously.

The results presented here are congruent with both the earlier-mentioned concept of a non-homogenous distribution of spores within the sample as well as a transient shedding pattern, and are in concordance with other equine longitudinal studies [46,47]. In one study, an overall *C. difficile* prevalence of 11% in hospitalised horses was observed; however, over a period of 7 months, no horse tested positive more than once [46]. These findings were consistent with a year-long study of healthy horses in which less than 10% of the subjects tested positive for the same *C. difficile* strain on more than one occasion [47]. These results were attributed to short-term stress factors, such as feed changes, transportation and seasonal changes, that may lead to a disruption of the natural gut flora. This ephemeral pattern has also been demonstrated in other animals and adds to the complexity of CDI epidemiology in animals [48].

The characterisation of *C. difficile* isolates is also a key component of CDI surveillance and epidemiological investigations. Overall, 64.2% of all strains isolated in this study were toxigenic and found across all population groups studied regardless of age, origin and GI health status. These toxigenic strains pose an ongoing threat of CDI within the horse population. The circulation of non-toxigenic strains of *C. difficile*, however, should not be overlooked. While incapable of causing symptomatic disease, the acquisition of the *C. difficile* PaLoc region (on which the genes for toxins A and B are located) by lateral gene transfer has been experimentally demonstrated with the potential threat of the emergence of toxigenic strains [49,50]. Furthermore, non-toxigenic strains have a protective function against toxigenic *C. difficile*—a characteristic currently being explored as a biotherapeutic for the prevention of CDI [51,52,53]. Non-toxigenic strains of *C. difficile* will often go undetected due to diagnostic testing methods based on toxin or toxin gene detection. Investigation of these strains may prove beneficial for ongoing surveillance to monitor for emerging and potentially useful strains. 

The need for surveillance is also exemplified in the large number of novel RTs of *C. difficile* isolated in this study, with 63 out of 148 isolates identified as new/unique strains. The high diversity of RTs is not surprising albeit problematic for characterisation. As investigation into community-acquired CDI gains momentum and research into animal and environmental sources of *C. difficile* continues, it is likely more novel RTs will be identified in these previously under-explored populations. All these factors highlight the need for a “One Health” approach to this problem, requiring input from both the medical and veterinary professions as well as environmental scientists.

Despite these novel findings, a majority of the isolates were recognised at both international and local levels. The most common of these, *C. difficile* RT 014/020, has the most prevalent RT of *C. difficile* in previous WA *C. difficile* studies of humans, pigs and public spaces [54,55]. Other commonly isolated strains within this study included *C. difficile* RTs 012, 010, 087 and 002, which are also prominent in humans, suggesting significant inter-species overlap [56,57,58]. Interestingly, RT 078, which dominates horse studies in the northern hemisphere [59,60], was not isolated in this study. Indeed, *C. difficile* RT 078 is rarely isolated in Australia, although the closely related RT 126 is [61].

The close association between the equine, human and environmental *C. difficile* ST54 (RT 012) strains seen in this study further supports the concept of inter-species transmission and interaction with the environment. *C. difficile* RT 012 was the most common RT in an early Australian equine *C. difficile* study of domesticated horses conducted from 2007 to 2009 [16], and is the second-most common here over a decade later. Using high-resolution whole-genome SNP analysis and applying estimates of the *C. difficile* molecular clock of ~1 SNP per genome per year [54,62], a genomic cluster of three horse strains and one human strain with clonal origins was identified, although common source exposure could not be ruled out. Previous genomic-based studies focused on *C. difficile* RT 014 have provided similar evidence for long-range inter-species transmission of *C. difficile* within Australia [54], suggesting that human and animal CDI are not occurring in isolation and further supporting a One Health paradigm for CDI.

This study highlights the benefit of genomics to investigate *C. difficile* epidemiology. As the reliance on genomics develops, genome-based research and the creation of a reference genome such as the one created here becomes increasingly important. To our knowledge, this is the first published study of the genomic epidemiology of equine CDI in Australia and is the first published reference genome of a strain of *C. difficile* isolated from a horse. Complete reference genomes create the basis for accurate phylogenetic and comparative analyses of *C. difficile* and can only enhance the existing knowledge pool of epidemiology, evolution and AMR.

The AMR in isolates from this study was more prevalent than that reported in an earlier Australian equine study [16]. Although susceptibility to CDI treatment agents remained high, with the emergence of broader AMR, including in non-toxigenic strains, the pool of resistance elements in circulating *C. difficile* is widening. AMR in non-toxigenic strains of *C. difficile* has been described in human population studies [63], and while this may not pose an immediate risk, the potential for the transfer of resistance genes to toxigenic strains of *C. difficile* and other infectious agents is of concern. Antimicrobial stewardship in veterinary medicine in Australia is lacking, with recent studies into veterinary practices in Australia showing that antimicrobial dispensing in equine and cattle without formal consultation was a common occurrence [64]. These findings warrant further investigation into antimicrobial use in animal sectors and its links to *C. difficile* in the wider community.

## 5. Conclusions

The data presented provide a sound knowledge base of the genomic epidemiology of *C. difficile* in domestic equine populations in WA. The diversity, AMR and overlap of strains with human isolates indicate that ongoing surveillance of *C. difficile* in Australian horse populations is warranted. The results presented here further support the hypothesis that horses represent a possible reservoir for *C. difficile* dispersal to humans, other animals and the environments and highlight the need for a holistic “One Health” approach to *C. difficile* research, prevention and control.

## Figures and Tables

**Figure 1 microorganisms-11-01743-f001:**
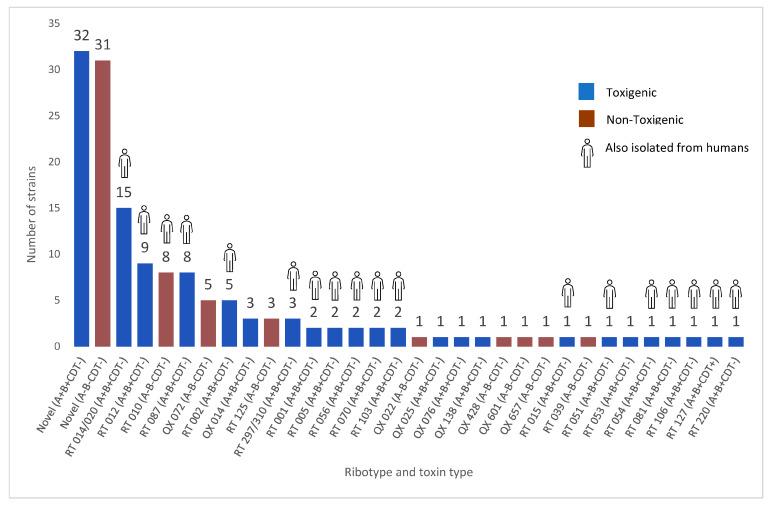
Characteristics of *C. difficile* isolated from both veterinary hospital and private domestic sources. Internationally recognised strains are prefixed “RT” and locally identified strains are prefixed “QX”. Toxigenic strains are those which contain at least one of the toxin genes *tcdA* (A+), *tcdB* (B+) or *cdtA/B* (CDT+).

**Figure 2 microorganisms-11-01743-f002:**
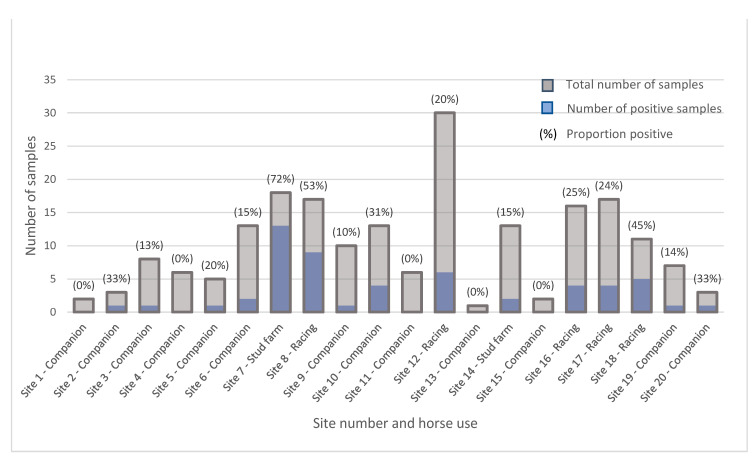
Proportions of equine faecal samples collected from Western Australian private domestic sources containing *C. difficile*. Sites categorised based on horse use include companion animals, racing training centres and stud farms.

**Figure 3 microorganisms-11-01743-f003:**
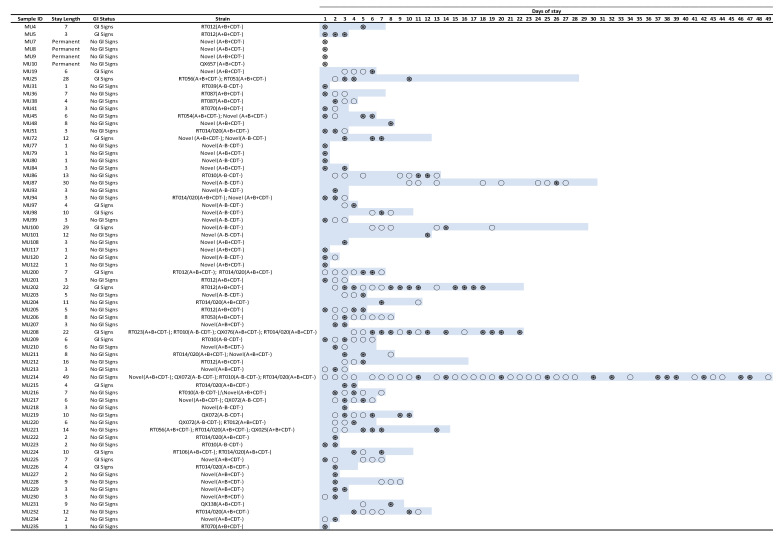
Isolation of *C. difficile* in patients at the TAHMU Equine Services over time. Duration of stay is indicated by the blue shading. Open circles indicate a negative sample, filled circles indicate a positive sample.

**Figure 4 microorganisms-11-01743-f004:**
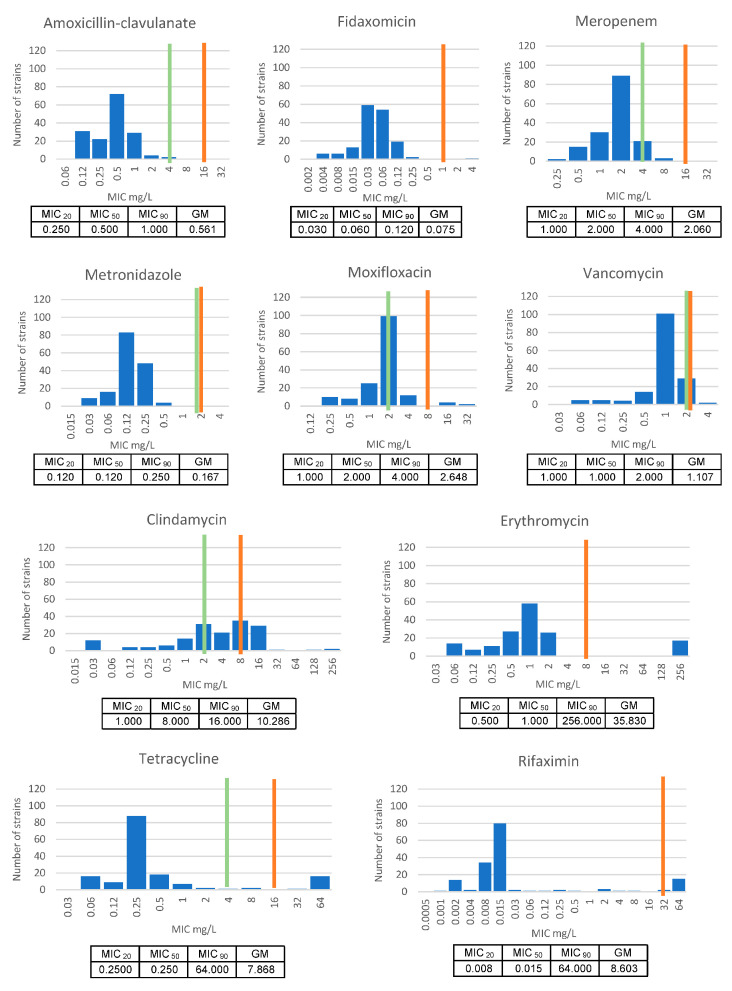
MIC distributions for 10 antimicrobial agents against 160 equine strains of *C. difficile*. Breakpoints are indicated with green (susceptible) and red (resistant) vertical markers.

**Figure 5 microorganisms-11-01743-f005:**
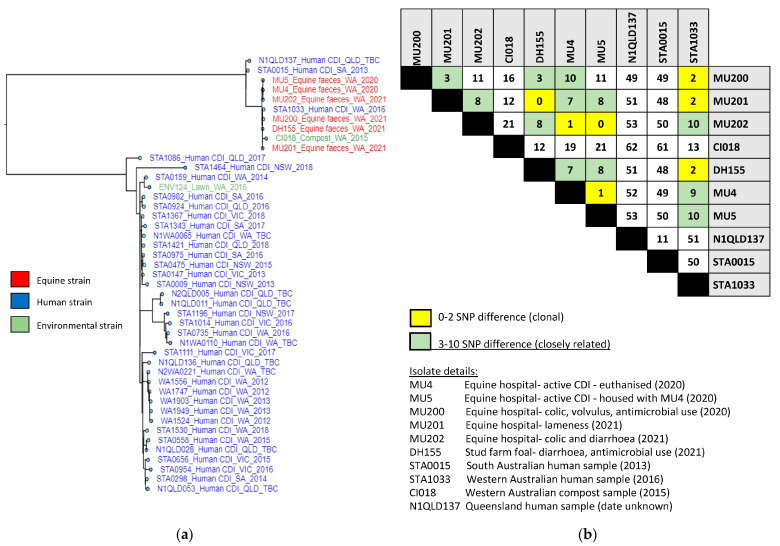
ST54 whole-genome SNP phylogeny. (**a**) Maximum-likelihood tree of 45 *C. difficile* PCR ribotype 012 (ST54) strains isolated from equine (*n* = 6), environmental (*n* = 2) and human origins (*n* = 37). Tree was built from an alignment of 1397 polymorphic sites and is mid-point-rooted. Scale bar shows nucleotide substitutions per site. (**b**) Pairwise whole-genome SNP analysis of ST54 cluster comprising equine (*n* = 6), clinical (*n* = 3) and environmental (*n* = 1) isolates.

**Table 1 microorganisms-11-01743-t001:** Proportion positive for *C. difficile* identified in the samples collected from veterinary hospital and private domestic sources including stud farms. GI = gastrointestinal. N/A = Not applicable.

Sample Origin	Horses (N)	*C. difficile* +ve Horses	Proportion +ve (%)
**Veterinary Hospital:**	**186**	**68**	**36.6%**
No GI signs	142	52	36.6%
Foals 4–12 months	0	n/a	n/a
Adults > 12 months	142	52	36.6%
GI signs	44	16	36.4%
Foals 4–12 months	0	n/a	n/a
Adults > 12 months	44	16	36.4%
**Private domestic:**	**201**	**55**	**27.4%**
No GI signs	194	52	26.8%
Foals 4–12 months	15	7	46.7%
Adults > 12 months	179	45	25.1%
GI signs	8	3	37.5%
Foals 4–12 months	5	3	60.0%
Adults > 12 months	3	0	0%
**Overall totals:**	**387**	**123**	**31.8%**
No GI signs	336	104	30.9%
Foals 4–12 months	15	7	46.7%
Adults > 12 months	321	97	30.2%
GI signs	51	19	37.2%
Foals 4–12 months	5	3	60.0%
Adults > 12 months	46	16	34.8%

**Table 2 microorganisms-11-01743-t002:** Features of a *C. difficile* RT 012 (ST 54) closed reference genome constructed by hybrid assembly.

Isolate MU4	Features
Origin	Domestic Horse, Western Australia, 2020
PCR ribotype	012
Multi-locus sequence type	54 (clade C1)
Toxin profile	A+B+CDT-
Chromosome	
GenBank accession	CP102397.1
Size (bp)	4,492,337
%GC	29.29
N CDS	4079
N tRNA/rRNA	50/6
N CRISPRs	9
Prophages	Clostridium Phage CDMH1 Clostridium Phage ΦMMP02 Clostridium Phage ΦCDHM19
No. genes	4309
No. coding sequences	4180
Noncoding RNAs	4
AMR loci	Beta Lactam (blaCDD-1) Vancomycin (vanR-Cd, vanS-Cd, vanG-Cd, vanT-Cd, vanZ1) Macrolide (erm(B)) Streptomycin (aadE, ant(6)-Ia, sat4) Tetracycline (tet(M)) Amikacin; kanamycin (aph(3’)-IIIa) Trimethoprim (dfrF) (95.96% coverage) Amikacin; gentamicin; kanamycin; tobramycin (aph(2”)-Ih) (82.21% coverage)
Extrachromosomal features	
Novel plasmid	7881 bp, accession CP102399.1
Novel phage	34,753 bp, accession CP102398.1
Total genome size (bp)	4,535,071
N contigs	3
Total ONT reads	978,529
Average ONT read length (bp)	3610
Basecalled N50	11,200
Total illumina reads	2,079,414
SRA accession	SAMN30090499

## Data Availability

Complete genome assembly data were submitted to GenBank under BioProject PRJNA865075 (accessions [CP102397, CP102398, CP102399]).

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
