# Peer review of "Genomic Analysis of Clostridioides difficile Recovered from Horses in Western Australia"

_microorganisms, 2023, doi:10.3390/microorganisms11071743_

Round 1
Reviewer 1 Report
Very professional scientific work. I have two questions.
1. Do you have the consent of the bioethics committee for the conducted research?
2. Are there cases of C. difficile being isolated from humans and animals with ribotype RT027 and RT176, A+B+CDT+ in Australia?
Strains with these ribotypes are hypervirulent and epidemic among human populations in Europe.
Reviewer 2 Report
In this study by Hain-Saunders and colleagues, the authors report the isolation and characterization of C. difficile from horses with and without enteritis. The authors supplemented this with the analysis of the whole genome of selected strains. In addition, the authors reported the resistance phenotype of the strains.
Overall, the article is well written and present a valuable knowledge about C. difficile in horses, for which the available reports are limited. Reading the manuscript shows that the approach they have taken is justified. I regret to admit that I have no major suggestions for improving the manuscript, only minor points that the authors may wish to consider.
Line 172: What software was used for MLST matching with the PubMLST database?
Line 176: Why is it necessary to create a separate custom database for these toxins when they already exist in the virulence factor database and can be used with abricate?
Line 184 - 185: This sentence needs to be revised as the iTOL is only a tool for visualization and not for creating a ML tree.
Lines 188 - 199: This text needs to be revised. The text mixes the analysis of short and long reads, e.g. filtlong is a specific tool for filtering long reads (not usually used for short reads). Why did the authors choose to use unicycler for the analysis, since the more recommended approach for assembling ONT data is flye, followed by medaka polishing, then polishing with the short reads.
Lines 198 - 199: This sentence describes AMR detection. In my opinion, this should be placed in section 2.5.
